

# Genome-wide analysis of basic helix-loop-helix transcription factors in papaya (*Carica papaya* L.)

Min Yang, Chenping Zhou, Hu Yang, Ruibin Kuang, Bingxiong Huang and Yuerong Wei

Guangdong Academy of Agricultural Sciences, Key Laboratory of South Subtropical Fruit Biology and Genetic Resource Utilization (MOA), Guangdong Province Key Laboratory of Tropical and Subtropical Fruit Tree Research, Institute of Fruit Tree Research, Guangzhou, China

## ABSTRACT

The basic helix-loop-helix (bHLH) transcription factors (TFs) have been identified and functionally characterized in many plants. However, no comprehensive analysis of the bHLH family in papaya (*Carica papaya* L.) has been reported previously. Here, a total of 73 *CpbHLHs* were identified in papaya, and these genes were classified into 18 subfamilies based on phylogenetic analysis. Almost all of the *CpbHLHs* in the same subfamily shared similar gene structures and protein motifs according to analysis of exon/intron organizations and motif compositions. The number of exons in *CpbHLHs* varied from one to 10 with an average of five. The amino acid sequences of the bHLH domains were quite conservative, especially Leu-27 and Leu-63. Promoter *cis*-element analysis revealed that most of the *CpbHLHs* contained *cis*-elements that can respond to various biotic/abiotic stress-related events. Gene ontology (GO) analysis revealed that *CpbHLHs* mainly functions in protein dimerization activity and DNA-binding, and most *CpbHLHs* were predicted to localize in the nucleus. Abiotic stress treatment and quantitative real-time PCR (qRT-PCR) revealed some important candidate *CpbHLHs* that might be responsible for abiotic stress responses in papaya. These findings would lay a foundation for further investigate of the molecular functions of *CpbHLHs*.

## INTRODUCTION

Since plants are unable to move, plant growth and development are regularly affected by abiotic and biotic stresses, which impair yields and result in losses to farmers. For better growth and development, plants have to make use of a series of physiological and biochemical processes in their responses to multiple abiotic stresses by regulating gene expression (*Agarwal et al., 2006*; *Feller et al., 2011*; *Pires & Dolan, 2010*). Previous studies have demonstrated that these physiological and biochemical mechanisms are more likely to be a polygenic cooperative defense response induced by various stresses, rather than the single response of a single gene (*Zhang, Creelman & Zhu, 2004*). Therefore, the traditional method of obtaining the stress tolerance of plants by modification a single

Corresponding author
Yuerong Wei, weiyuerong@gdaas.cn

resistance/sensitive gene is limited. Comprehensive analysis of important gene families are very important for molecular breeding.

As an important and popular fruit, papaya is famous for its high nutritional and medical values. Papaya is widely grown in southern China, the tropics and subtropics areas, and its demand is increasing every year. However, the production and quality of papaya were often threatened by various abiotic stresses, such as salt, drought, and cold. These stresses often cause severe economic losses in papaya production in China. Therefore, it is very important to study the functions of gene families that involved in abiotic stresses response in papaya. Since obtaining the whole genome sequences of papaya (*Ming et al., 2008*), several important gene families have been identified by the tool of genome-wide analysis in papaya, including *Aux/IAA* gene family, ARF family, *SQUAMOSA promoter binding protein-like* (*SPL*) gene family, NBS resistance gene family and NPR1 family. These families were essential for papaya fruit ripening, flower and fruit development, fitness and disease resistance (*Liu et al., 2017a*; *Liu et al., 2015*; *Peraza-Echeverría, Santamaría & Fuentes, 2012*; *Porter et al., 2009*; *Xu et al., 2020*).

In various stresses regulation network and signaling pathways, transcription factors (TFs) are important proteins that regulate gene expression by activating and repressing related downstream genes. Among them, WRKY and bHLH families are the most common TF families in higher plants (*Kosugi & Ohashi, 2002*). The WRKY transcription factors has been reported to be related to abiotic and biotic stresses responses in papaya (*Pan & Jiang, 2014*). Basic/helix-loop-helix (bHLH) TFs are widely found in almost all eukaryotes and are the second largest TFs family in plants (*Carretero-Paulet et al., 2010*; *Feller et al., 2011*; *Jones, 2004*; *Pires & Dolan, 2010*). The bHLH superfamily proteins are defined by one highly conserved bHLH domain, which comprises approximately 60 amino acids in length and contains two different functional regions: the basic region and the HLH region (*Li et al., 2006*; *Toledo-Ortiz, Huq & Quail, 2003*). The basic region is located at the N-terminal end of the bHLH domain and consists approximately 15 amino acids. It is a DNA-binding region that enables bHLH TFs to bind to a specific E-boxes (CANNTG) (*Atchley & Fitch, 1997*; *Atchley, Terhalle & Dress, 1999*). The HLH region, at the C-terminal end, is mainly composed of hydrophobic residues, containing two amphipathic $\alpha$-helices linked by a loop region that has variable sequences and acts as a dimerization domain (*Heim et al., 2003*; *Li et al., 2006*). Outside of the two conserved regions, the rest of the bHLH protein sequences are usually very different (*Morgenstern & Atchley, 1999*).

In animals, the bHLH TFs can be divided into six main groups (designated A to F) based on phylogenetic analysis, functional properties and DNA-binding specificity (*Atchley & Fitch, 1997*). These bHLH groups can be divided into several small subfamilies (*Ledent & Vervoort, 2001*; *Simionato et al., 2007*). The bHLHs mainly function in sensing the external environment, cell cycle regulation and tissue differentiation (*Amoutzias, Robertson & Bornberg-Bauer, 2004*; *Atchley & Fitch, 1997*; *Stevens, Roalson & Skinner, 2008*; *Vervoort & Ledent, 2001*). Compared to animals, the research on bHLH proteins in plants is limited, even the exact number subfamilies of bHLH TFs has not been determined. Generally, the bHLH proteins is thought to cover 15–25 subfamilies (*Buck & Atchley, 2003*; *Pires & Dolan, 2010*), but some atypical bHLHs have extended the number to 32 based

on phylogenetic analysis in plants (*Carretero-Paulet et al., 2010*). With the availability of genome sequence data and the rapid development of molecular biology, increasing numbers of bHLH subfamily genes have been identified and characterized in a wide range of plant species, including *Arabidopsis* (*Toledo-Ortiz, Huq & Quail, 2003*), peanut (*Gao et al., 2017*), apple (*Mao et al., 2017*), tomato (*Sun, Fan & Ling, 2015*), potato (*Wang et al., 2018b*), peach (*Zhang et al., 2018*), grapes (*Wang et al., 2018a*), sweet orange (*Geng & Liu, 2018*), and bamboo (*Cheng et al., 2018*). The results from these research have shown that bHLH TFs have versatile biological functions, such as regulating light morphogenesis (*Leivar et al., 2008*; *Roig-Villanova et al., 2007*), hormone signals (*Friedrichsen et al., 2002*; *Lee et al., 2006*), the developmental of root (*Feng et al., 2017*) and anther (*Farquharson, 2016*), regulating epidermal cell fate determination (*Bernhardt et al., 2003*), participating in various biotic and abiotic stress responses (*Jiang, Yang & Deyholos, 2009*; *Liu et al., 2014*; *Wang et al., 2018b*), etc.

In recent years, some studies demonstrated that bHLH transcription factors play important roles in the stress-related regulation network and signaling pathways in many species. However, no systematic analysis of the bHLH TFs have previously been performed in papaya. In this study, a total of 73 *CpbHLH* genes were identified in papaya, and phylogenetic analyses were carried out to analyze the relationships among these genes. Meanwhile, gene structure, protein physicochemical properties and conserved motifs, the *cis*-element of the promoter region, and gene ontology (GO) analysis were investigated. Furthermore, to analyze the functions of *Cp*bHLHs responsible for responding to abiotic stresses, the expression profiles of 22 selected genes under salt, drought, ABA and cold stresses were investigated by using quantitative real-time PCR (qRT-PCR). We identified several important candidate genes that might be responsible for abiotic stress responses. We completed the first comprehensive genome-wide analysis of the *bHLH* gene family in papaya, and our results provide information necessary for further functional research of the bHLH family in papaya.

## MATERIALS & METHODS

### Identification of *CpbHLH* genes, gene structure and physicochemical analysis

Papaya (*Carica papaya* L.) bHLH protein sequences were downloaded from the Plant TFDB V4.0 database (*Jin et al., 2017*). Furthermore, we used the SMART online software (http://smart.embl-heidelberg.de/) and the InterProScan tool (http://www.ebi.ac.uk/Tools/pfa/iprscan/) to identify integrated bHLH domains in putative papaya bHLH proteins. The physicochemical properties of *Cp*bHLH proteins were predicted by ProPAS (*Wu & Zhu, 2012*). The genomic sequences, ID numbers and coding sequences (CDS) corresponding to each predicted *CpbHLH* gene were obtained from the Phytozome database (https://phytozome.jgi.doe.gov/pz/portal.html). The intron numbers, exon–intron organizations and locations of the *CpbHLH* genes were analyzed by Gene Structure Display Server (GSDS) v2.0 (*Hu et al., 2015*).

## Phylogenetic tree building, motif identification and multiple sequence alignment

To research the phylogenetic relationship of *Cp*bHLH proteins, protein sequences of papaya were pre-aligned using HMM align (*Eddy, 1998*) and the pHMM HLH ls.hmm from PFAM (https://pfam.xfam.org/family/PF00010) to identify the domains of bHLH TFs. Based on the manually aligned bHLH region of 158 bHLH proteins from *Arabidopsis* and 173 from rice (*Pires & Dolan, 2010*), the identified bHLH domains were later aligned using MAFFT v7.305b (*Kaotoh, Kuma & Miyata, 2002*) with default settings. Phylogenetic tree was constructed based on the neighbor-joining method using FastTree v2.1.11 (*Price, Dehal & Arkin, 2009*) with default settings. Bootstrapping with 1000 replicates was used to assess the statistical reliability of nodes in the tree. Multiple sequence alignment based on protein sequences of these 73 *Cp*bHLH TFs was generated by MAFFT v7.305b (*Kaotoh, Kuma & Miyata, 2002*) with default settings.

To identify the conserved motifs among the *Cp*bHLH proteins, we uploaded the 73 amino acid sequences of the *Cp*bHLH family to the Multiple EM for Motif Elicitation (MEME, version 5.1.1) (http://meme-suite.org/tools/meme). The parameter settings were as follows: zero or one, occurrence of a single motif per sequence; 3, maximum number of motifs found. All other parameters were set to the default values.

## Promoter *cis*-acting regulatory element analysis and gene ontology (GO) annotation

To predict and compare the putative promoter *cis*-elements of *bHLHs* in papaya and *Arabidopsis*, the upstream 2,000 bp genomic DNA sequences of 73 *CpbHLH* genes in papaya, and 47 *AtbHLH* genes in *Arabidopsis* (the putative orthologous genes corresponding to *CpbHLHs*) were downloaded and then submitted to the PlantCARE (*Magali et al., 2002*). The GO annotations of papaya bHLH proteins were downloaded from AgriGOv2.0 (http://systemsbiology.cau.edu.cn/agriGOv2/download/508_slimGO). For GO enrichment analysis, the full-length protein sequences of papaya bHLH were blasted against Arabidopsis proteins with default parameters. The best hits were submitted to AgriGOv2.0 for GO enrichment analysis (*Tian et al., 2017*). Fisher's exact test was used to evaluate enriched GO terms. GO terms include three aspects: biological process, cellular component and molecular function.

## Plant materials, growth conditions and stress treatments

In this experiment, stems with axillary buds were selected as explants from two-year-old 'Yi Chi Gua' papaya trees grown under standard field conditions in the Institute of Fruit Tree Research, Guangdong Academy of Agriculture Science, Guangzhou, China, and cultured in vitro to obtain the complete papaya seedlings with normal leaves and roots using tissue culture techniques. Healthy and uniform papaya seedlings were used for different treatments. For the selection of stress conditions for papaya, we designed different gradients of stress conditions for pre-experiments: the concentration gradients of salt stress are 100 mM Nacl, 200 mM Nacl, and 300 mM Nacl; the concentration gradients of PEG6000 (to mimic drought stress) are 15% PEG6000, 20% PEG6000, 25% PEG6000 and 30% PEG6000; the concentration gradients of ABA are 50 μM ABA, 100 μM

ABA and 150 μM ABA; the temperature gradients are 0 °C, 4 °C and 10 °C, and finally determined the suitable stress conditions used in this manucript. For salt, drought and ABA stresses, seedlings were treated with MS liquid medium containing 200 mM Nacl, 25% PEG6000 and 100 μM ABA for 2 h respectively, and then the roots were collected. For cold treatment, seedlings were subjected to 4 °C for 2 h and the leaves were collected. All of the collected materials were immediately frozen in liquid nitrogen and stored at −80 °C for RNA isolation. Untreated seedlings were used as the control groups. Three biological replications were carried out for each treatment.

### RNA extraction and quantitative real-time PCR (qRT-PCR) analysis

Total RNA from papaya after different treatments was isolated using TRIzol reagent (Invitrogen). The extracted RNA was treatment with DNase (TaKaRa), and then reverse transcribed into cDNA using the PrimeScriptTM RT Reagent Kit (TaKaRa). The qRT-PCR was conducted on the ABI StepOne Real Time PCR system using 2X SG Fast qPCR Master Mix (High Rox) (TaKaRa) according to the manufacturer's instructions. TATA binding protein 2 (*Cp*TBP2) amplification was used as an internal control (*Zhu et al., 2012*). The qRT-PCR reactions used three biological replicates, and each biological repeat had three technical replicates. Gene-specific primers for qRT-PCR of the 22 *CpbHLH* genes were designed based on the CDSs of the *CpbHLH* genes using Primer Premier 5.0 (Data S1). The relative expression levels of each gene were calculated using the $2^{-\Delta\Delta CT}$ method, the raw data was showed in File S1.

## RESULTS

### Identification and characterization of *Cp*bHLHs

A total of 105 putative bHLH transcription factors of papaya (*C. papaya*) were downloaded from the PlantTFDBv2.0 (http://planttfdb.cbi.pku.edu.cn/). To verify the reliability of these results, the 105 *Cp* bHLH proteins sequences were filtered by Interproscan and SMART domain annotation, and a total of 73 predicted *Cp*bHLH proteins were identified. They were named *Cp*bHLH001 to *Cp*bHLH073 at random except for 32 proteins that were explicitly excluded by Interproscan and SMART (Table S1). The detailed information on these predicted *Cp* bHLHs, including protein ID, locus ID, opening reading frame (ORF) lengths, amino acid sequences/lengths, molecular weight, isoelectric point and exon/intron numbers, are listed in Data S2. In previous studies, 129/132, 188, 159, 147, 124, 95, 94 and 56 *bHLH* genes were identified in peanut (*Gao et al., 2017*), apple (*Mao et al., 2017*), tomato (*Sun, Fan & Ling, 2015*), *Arabidopsis* (*Toledo-Ortiz, Huq & Quail, 2003*), potato (*Wang et al., 2018b*), peach (*Zhang et al., 2018*), grapes (*Wang et al., 2018a*) and sweet orange (*Geng & Liu, 2018*), respectively. Compared with the above dicotyledonous plants, the density of *bHLHs* genes in papaya genome was about 0.26%, which is lower than the density of peanut (*Gao et al., 2017*), apple (*Mao et al., 2017*), tomato (*Sun, Fan & Ling, 2015*), *Arabidopsis* (*Toledo-Ortiz, Huq & Quail, 2003*), potato (*Wang et al., 2018b*) and sweet orange (*Geng & Liu, 2018*), and similar to peach (*Zhang et al., 2018*) and wine grapes (*Wang et al., 2018a*) (Table 1). This is probably associated with the whole-genome duplications during evolution. Among the above plants, some plants with recent whole-gene duplication like

**Table 1  Summary of TFs identified from dicotyledonous plant species with genome sequences.**

| Plant species | Common name | bHLH | Proteins | Ratio (%) |
|---|---|---|---|---|
| *C. papaya* | Papaya | 73 | 27829 | 0.26 |
| *A. ipaensis/A. duranensis* | Peanut | 129/132 | 7243 | 1.78/1.82 |
| *Malus x domestica* | Apple | 188 | 15173 | 1.24 |
| *S. lycopersicum* | Tomato | 159 | 15722 | 1.01 |
| *A. thaliana* | Arabidopsis | 147 | 32125 | 0.46 |
| *S. tuberosum* | Potato | 124 | 17445 | 0.71 |
| *P. persica* | Peach | 95 | 28299 | 0.34 |
| *V. vinifera* | Wine Grapes | 94 | 47097 | 0.20 |
| *C. sinensis* | Valencia Orange | 56 | 13522 | 0.41 |

peanut, apple, tomato, Arabidopsis, potato and sweet orange while the plants without whole-gene duplication like papaya, peach and wine grapes.

To further characterize the bHLHs in papaya, the physicochemical properties of these putative proteins were analyzed and are shown in Data S2. The size of deduced *Cp*bHLHs ranged from 100 (*Cp*bHLH053) to 679 (*Cp*bHLH068) amino acids, the corresponding molecular weights from 11.525 KDa to 75.899 KDa. The predicted theoretical isoelectric points (PI) values of *Cp*bHLHs were between 4.71(*Cp*bHLH028) and 11.07(*Cp*bHLH003). Similar molecular weights and isoelectric points have been made in potato (*Wang et al., 2018b*). And all of predicted *Cp*bHLH proteins were hydrophilic characteristic proteins, the grand average of hydropathy values were negative, ranging from −0.2098(*Cp*bHLH033) to −1.0125(*Cp*bHLH006). Similar result has been made in *Brachypodium distachyo* n (*Niu et al., 2017*). That is, the predicted *Cp*bHLH proteins showed diversities in their length, molecular weight, PI and the grand average of hydropathy values.

## Phylogenetic analysis, gene structure, conserved motifs analysis and multiple sequence alignment of *Cp*bHLHs

To evaluate the evolutionary relationships of the *Cp*bHLH proteins, a neighbor-joining phylogenetic tree was generated using conserved bHLH domains from papaya, *Arabidopsis* and rice. The phylogenetic tree showed that the 73 *Cp*bHLH members were clustered into 18 subfamilies with one orphan (Fig. 1A and Data S3), consistent with the earlier results showing that the bHLH subfamily in plants can be divided into 15–25 subfamilies (*Pires & Dolan, 2010*). Previous research have named the bHLH subfamilies using English letters (*Li et al., 2006*; *Mao et al., 2017*), Roman numerals (*Song et al., 2017*; *Sun, Fan & Ling, 2015*), or Arabic numerals (*Chen et al., 2015*; *Toledo-Ortiz, Huq & Quail, 2003*), In this study, we named *Cp*bHLH subfamilies using Roman numerals. As shown in Fig. 1, the subfamily XII was the largest subfamily among all three species, and all of subfamilies include at least two species. In papaya, none of the bHLHs were grouped into IV d, II, X V, X, XI V and XIII subfamilies compared to rice and *Arabidopsis*, which may be due to these bHLHs were lost during the process of evolution.

Exon/intron organization, as a type of structural divergence, plays an important role in the evolution of multiple gene families (*Xu et al., 2012*). The annotation features of the

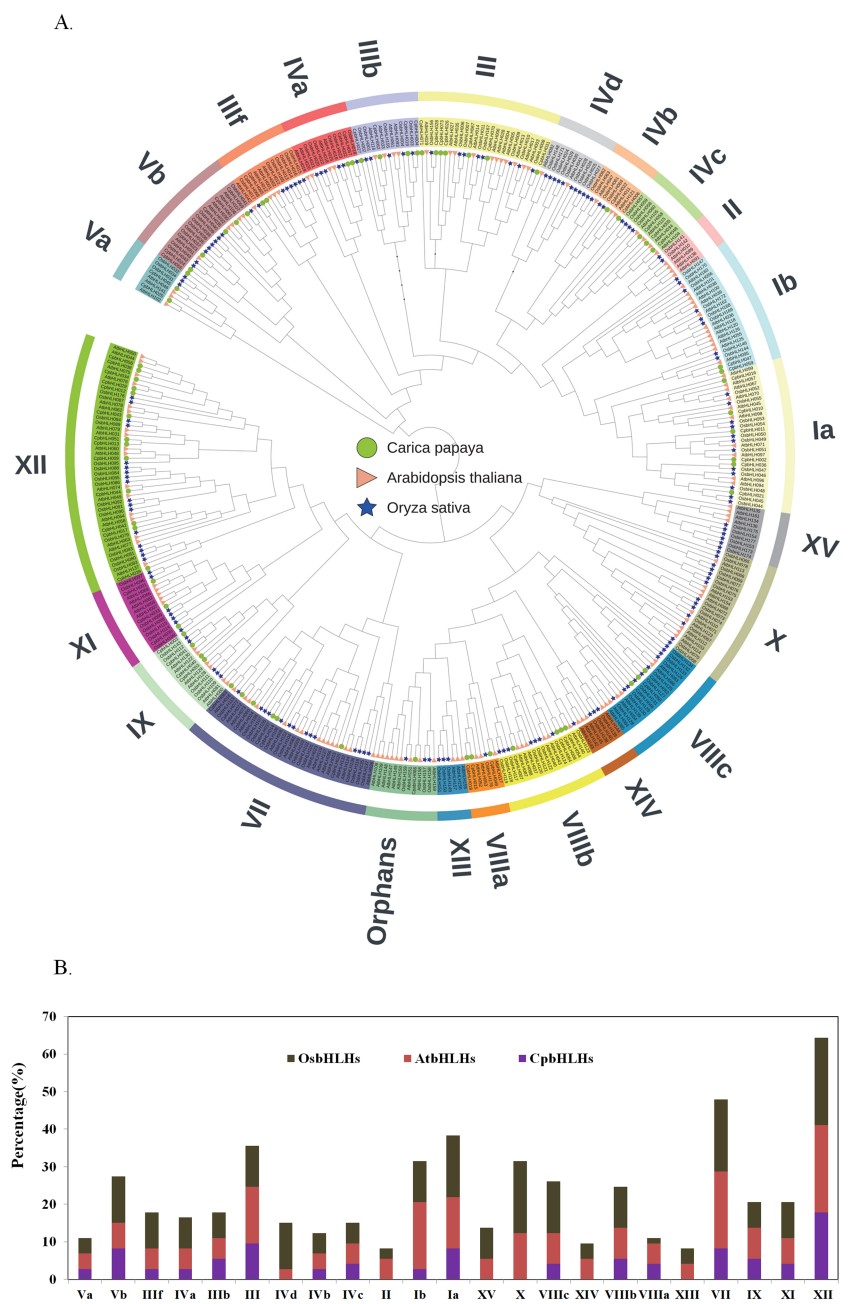

**Figure 1 Phylogenic and family members analysis of bHLHs from papaya, rice and *Arabidopsis*.**
(A) The 73 *Cp*bHLHs are clustered into 18 subfamilies. Phylogenetic tree was constructed based on the neighbor-joining method. Bootstrapping with 1,000 replicates was used to assess the statistical reliability of nodes in the tree. (B) Comparison of bHLH family members from papaya, rice and *Arabidopsis*. Different colors represent the different plants. Black: *Os*bHLHs, red: *At*bHLHs, purple: *Cp*bHLHs.

*CpbHLH* genes were submitted to Gene Structure Display Server (GSDS) together to show their gene structures. As described in Data S2 and Fig. 2A, the number of introns varied from zero to ten, representing a complex distribution pattern. Most (63 (86.3%)) of the

Peer**J**

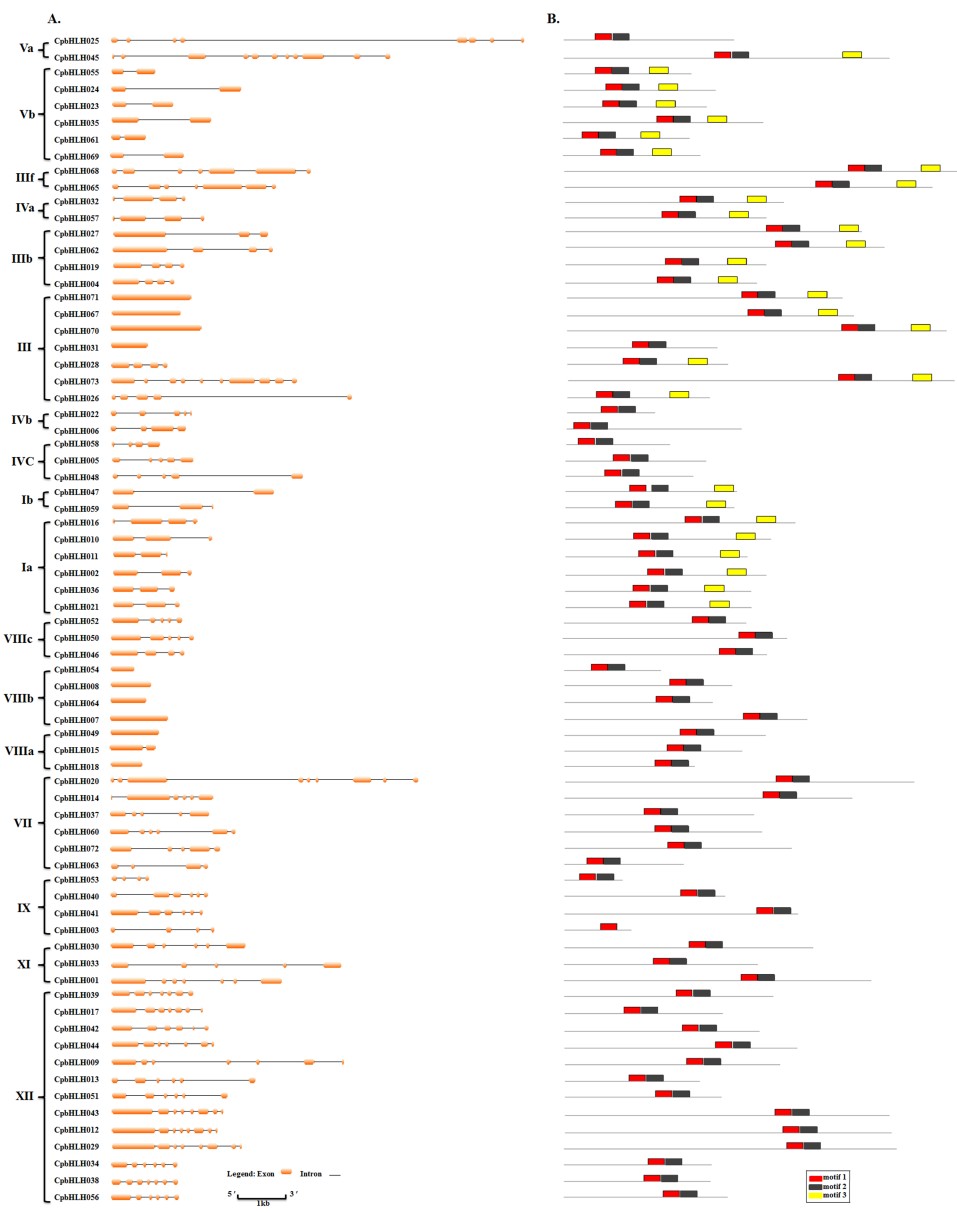

**Figure 2** **Gene structure and motif distribution of the papaya bHLH family.** (A) Exon-intron organization of *CpbHLH* genes. Exons and introns are presented as filled orange lines and thin black single lines, respectively. The brackets and Roman numerals separate each subfamily and clearly present the member conservation of each subfamily. (B) Arrangements of conserved motifs in 73 *Cp*bHLH proteins. Three predicted motifs are represented by different colored boxes, motif 1 (red block), motif 2 (black block) and motif 3 (yellow block).

*CpbHLHs* were found to possess introns among the 73 *CpbHLH* genes, while 10 (13.7%) of the genes were intron-less, 8 (11.0%) genes contained one intron, and the remaining genes had two or more introns. In addition, members of the same subfamily also displayed similar intron distribution patterns in view of the full-length genome sequences. For instance, all of the *CpbHLHs* in subfamily V b had one intron and two exons, the whole members of

subfamily III f had six introns and seven exons, the IV a subfamily members showed three introns and four exons, and all members of VIII b subfamily consisted only one exon.

Most importantly, members of the same bHLHs subfamily are usually participated in the same signaling pathway or biological process, and the functions of these members are often partially or totally redundant (*Pires & Dolan, 2010*). For example, *At*bHLH10, *At*bHLH89 and *At*bHLH91, corresponding rice orthologs *Os*bHLH141, *Os*bHLH142 are members of subfamily II, they are all involved in the process of pollen development (*Li et al., 2006*; *Liu et al., 2017*; *Zhu et al., 2015*). Especially in *Arabidopsis*, there is no obvious phenotype in single mutant of *AtbHLH10*, *AtbHLH89* or *AtbHLH91*, only their various double or triple mutants showed the phenotype of pollen development deficiency (*Liu et al., 2017*). In subfamily III b, *Os*bHLH001 (*OsICE2*), *Os*bHLH002 (*OsICE1*), *Cp*bHLH027, *Cp*bHLH062, *At*bHLH116 (ICE1) and *At*bHLH33 (ICE2) were clustered within one clade. In previous studies, *At*bHLH116(ICE1) and *At*bHLH33(ICE2) and corresponding orthologs in rice (*Os* bHLH001/*Os*ICE2, *Os*bHLH002/*Os*ICE1) have been reported to function in the stress of chilling (*Chinnusamy et al., 2003*; *Deng et al., 2017*; *Fursova, Pogorelko & Tarasov, 2009*; *Li et al., 2010*; *Zhang et al., 2017*). And we also found transcripts of *Cp*bHLH027 and *Cp*bHLH062 were increased under chilling stress in this study, implying that *Cp*bHLH027 and *Cp*bHLH062 are involved in the process of chilling stress in papaya.

To further study the sequence characteristics of the predicted bHLH domains at the amino acid level, we carried out a multiple sequence alignment of the 73 predicted *Cp*bHLH protein sequences (Fig. 3). The result showed that the 73 putative *Cp*bHLH proteins contained two conserved regions in the bHLH domains: the basic region plus helix 1 and the loop region plus helix 2 (Fig. 3 and Table S2). Additionally, we used the online MEME program to identify the conserved motifs (*Bailey & Elkan, 1994*). The result also showed that most of the sequences (exclude *Cp*bHLH003) exhibited two highly conserved motifs: one is contains 29 amino acids, and the other consists of 21 amino acids, are shown in red and black blocks, respectively (Fig. 2B). Among the two motifs, motif 1 comprises basic residues and helix 1, and motif 2 comprises a loop and helix 2. And the space between motif 1 and 2 consists of a loop, which is variable in length in some bHLH proteins. The sequence logos of motif 1 (in red) and motif 2 (in black) are shown in Fig. 4A. The backbones of motif 1 and 2 are also conserved in most plant species (*Guo & Wang, 2017*; *Heim et al., 2003*; *Sun, Fan & Ling, 2015*), and these highly conserved residues in bHLH domains may be responsible for protein dimerization (*Heim et al., 2003*).

Besides these two common conserved motifs, some *Cp*bHLHs that are mainly distributed into eight subfamilies (including V a, V b, III f, IV a, III b, III, I b and I a subfamilies) harbor another conserved motif (motif 3) with a length of 36 amino acids. The motif 3 is indicated by the yellow blocks and the sequence logo is visualized as logo3 (Fig. 2B and Fig. 4B). This result is accord with the previous studies that members of a given subfamily exhibited another conserved nonbHLH motif (motif 3) in plant bHLH superfamily (*Pires & Dolan, 2010*). However, in papaya, members of the bHLH proteins have the same motif that is distributed into eight subfamilies, not just one subfamily. In addition, among the 73 *Cp*bHLHs, one atypical bHLH protein (*Cp* bHLH003) exhibited incomplete bHLH domains, whereas the remaining 72 *Cp*bHLH proteins all presented complete bHLH

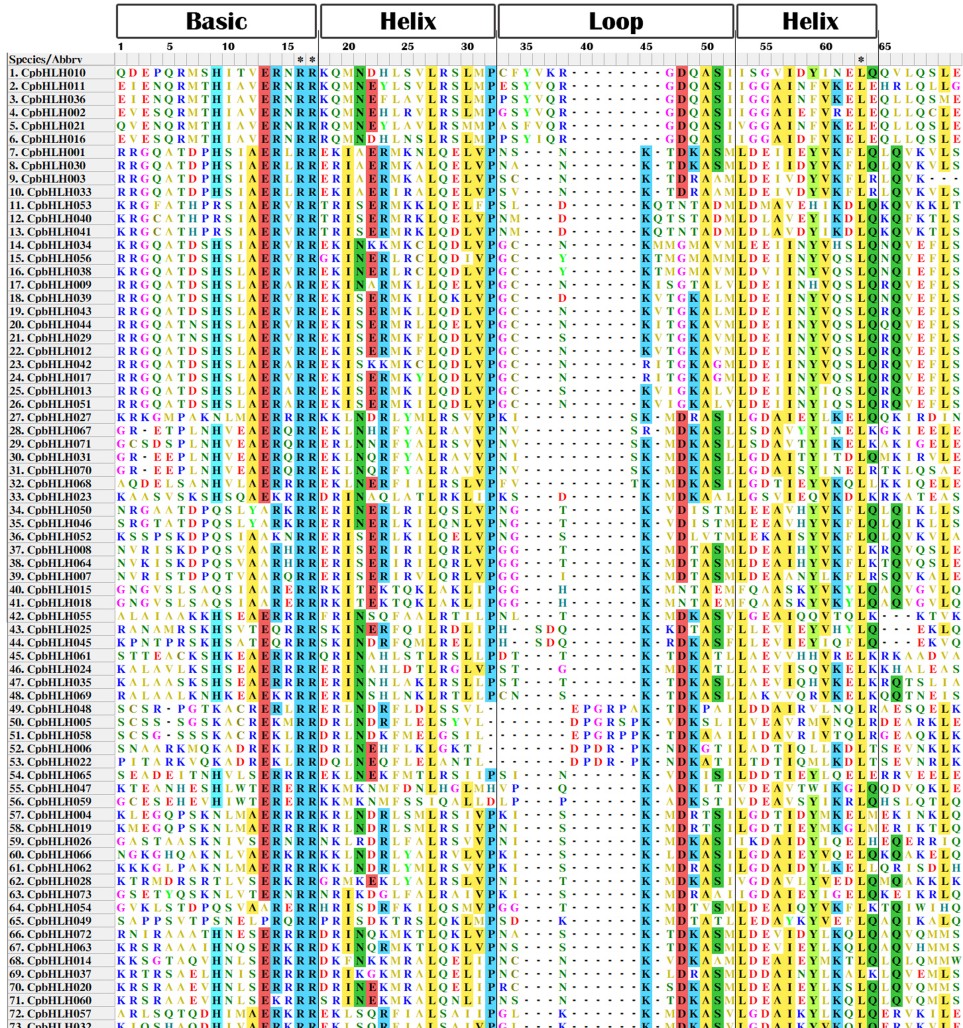

**Figure 3   Multiple sequence alignment of the bHLH domains in papaya.** Amino acids with more than 50% identity are labeled with colored boxes.

domains. Similar observations have been made in other plant species, such as peach and blueberry (*Song et al., 2017*; *Zhang et al., 2018*).

## Promoter analysis of *bHLH* genes in papaya

To further understand *Cp*bHLHs functions and regulation patterns, *cis*-elements in *CpbHLH* genes promoter sequences were investigated. Regions of 2,000 bp upstream from the start codons of each *CpbHLH* gene were analyzed using PlantCARE. The results showed that the *cis*-elements could be divided into three main categories (Fig. 5A and Data S4). Category one contained a ubiquitous class of plant light responsive elements among which G-Box, G-box, GT1-motif and Box 4 were common in the *CpbHLH* promoters. Category two contained important elements that were involved in the process of stress-responsiveness, including MYB binding site involved in drought-inducibility
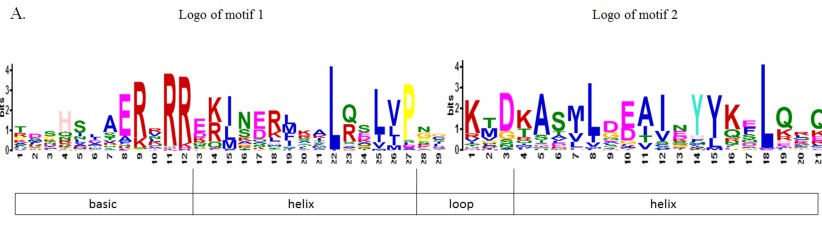

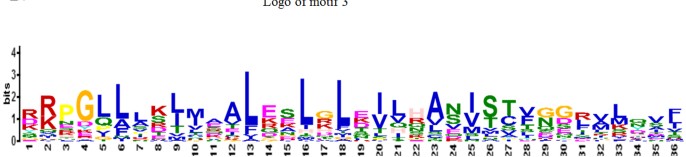

**Figure 4** **Motif composition and logos of papaya bHLH proteins.** (A) The logos of motif 1 and 2, which together constitute the bHLH domain in papaya. The overall height of the character represents the conservation of an amino acid at the specific position. Each color of the English letters represents a type of amino acid residue. (B) The logo of motif 3, which is another conserved motif.

(MBS), low temperature response elements (LTR), defense and stress responsive elements (TC-rich) and wound-responsive elements (WUN-motifs). In addition, more than ten kinds of hormone-responsive *cis*-elements were identified (e.g., gibberellin-GA, auxin-IAA, methyl jasmonate-MeJA, salicylic acid-SA, and abscisic acid-ABA). Among them, the most common response elements were ABA (ABRE), MeJA (CGTCA-motif and TGACG-motif) and SA (TCA-element and SARE), which included 158 (29.15%), 128 (23.62%) and 53 (9.78%), respectively (Fig. 5B). Category three contained plant growth and development elements, such as anaerobic induction elements (ARE), $O_2$-site, CAT-box and so on. Additionally, we also analyze the *cis*-elements in the promoter regions of putative orthologous genes that corresponding to *CpbHLH* in *Arabidopsis*, and the similar result has been obtained in *Arabidopsis* (Fig. S1 and Data S4). There also existed three main categories: plant light, abiotic and biotic stresses and plant growth and development responsive elements. And the percentage of most stress-responsive elements in *Arabidopsia* were similar to papaya, including ABA responsive elements, drought-responsive elements, wound-responsive elements, low temperature-responsive elements and IAA responsive elements, implying that most of the promoter *cis*-elements of bHLH family were conserved in *Arabidopsis* and papaya.

## GO annotation of *Cp*bHLH proteins

To understand the functions of papaya bHLHs, we performed a GO annotation of *Cp*bHLHs, and the results are shown in Data S5 . A total of 70 *Cp*bHLHs were involved in protein dimerization activity (GO: 0046983). The result is consistent with the earlier studies, which show that the HLH domain was necessary for protein dimerization and DNA binding (*Murre, Mccaw & Baltimore, 1989*). Some conserved amino acid residues are important to the function of bHLH proteins, especially the Leu-27 in helix 1 and the Leu-73 in helix

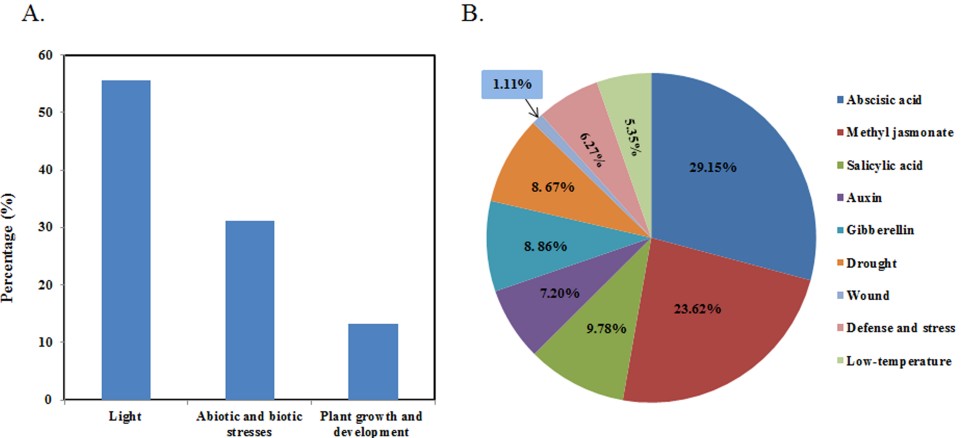

**Figure 5** *Cis*-acting element analysis of the promoter of *bHLH* genes in papaya. (A) Percentage of total *cis*-acting elements in the promoter region of *CpbHLH* genes. (B) The percentage of each *cis*-acting element in the abiotic and biotic stresses categories. Abscisic acid (29.15%), methyl jasmonate (23.62%), salicylic acid (9.78%), auxin (7.20%), gibberellin (8.86%), drought (8.67%), wound (1.11%), defense and stress (6.27%), low temperature (5.35%).

2 (*Carretero-Paulet et al., 2010*). In this study, we found 72 (out of 73) *Cp*bHLH proteins have Leu-27 (corresponding to Leu-27 in *At*bHLHs), and all of the *Cp*bHLH proteins have Leu-63 (corresponding to Leu-73 in *At*bHLHs) (Fig. 3 and Table S2). Because of a lack of reported experimental data and databases, we used *Arabidopsis* as the reference species to perform a GO enrichment analysis of *Cp*bHLH proteins, and 54 of 73 predicted *Cp*bHLH proteins were obtained with results compared to *Arabidopsis*. We summarized the results in Fig. 6 and Data S6. The majority of predicted *Cp*bHLH proteins were enriched in DNA binding. Almost all of the predicted *Cp*bHLH proteins (37, 68.5%) were predicted to localize in the nucleus, whereas the remaining predicted *Cp*bHLH proteins were located in other organelles, including plastids, the cytoplasm, and chloroplasts. Additionally, some predicted *Cp*bHLH proteins existed in multiple cellular components. For example, *Cp*bHLH013 was located in three cellular components: chloroplasts, part of the cytoplasm, and the nucleus, which may reflect its multiple functions in various biological processes. The metabolic processes involved the greatest number of putative *Cp*bHLH proteins (47, 87.0%). Biosynthetic processes and gene expression involved the second greatest number of putative *Cp*bHLH proteins (46, 85.2%). In addition, *Cp*bHLH proteins could respond to stimulus, morphogenesis, cell differentiation, and developmental process.

## Expression analysis of bHLH superfamily genes under different abiotic stresses

The bHLH proteins have been characterized functionally in many plants with a vital role in the regulation of diverse biological processes, but little is known about their role in papaya. To analyze the functions of *Cp*bHLHs responding to abiotic stresses, the expression profiles of 22 selected genes under salt, drought, ABA and cold stresses were investigated by using qRT-PCR (Table 2 and Fig. 7). The results showed that 4 (*CpbHLH011*, *CpbHLH022*,

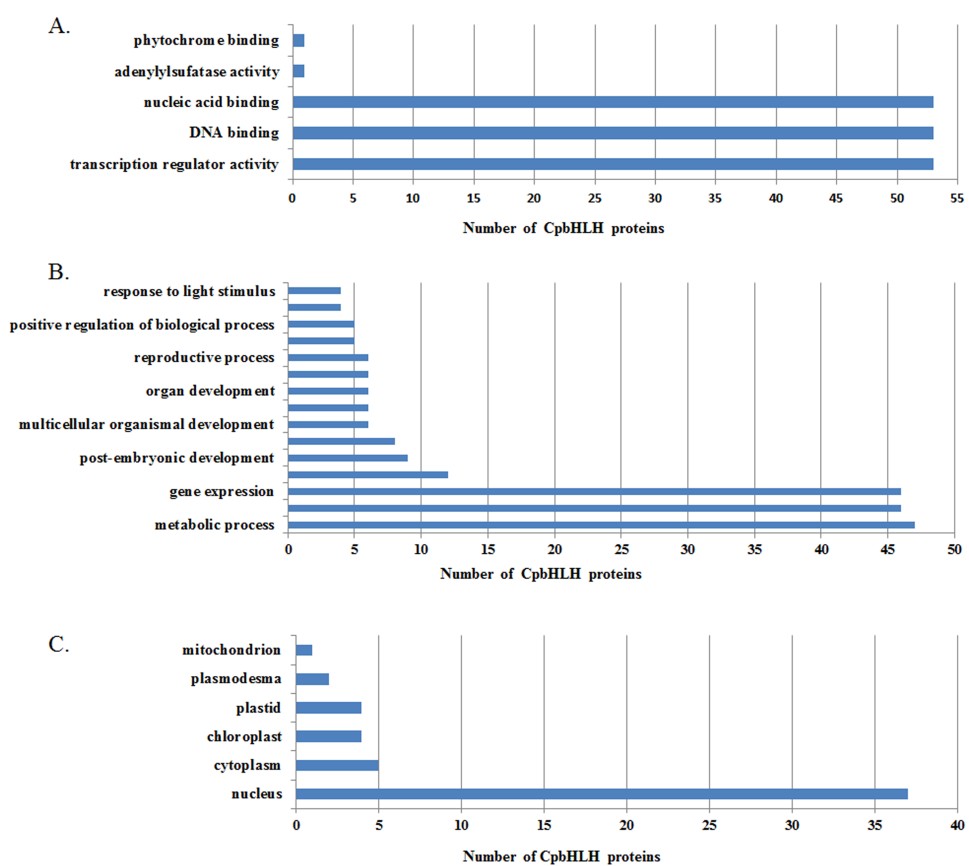

**Figure 6  Gene ontology (GO) annotation of *Cp* bHLH proteins.** The annotation was performed on three categories: (A) molecular function, (B) biological processes and (C) cellular components.

*CpbHLH027* and *CpbHLH056*) of 22 *Cp*bHLH mRNAs were increased, and 3 *CpbHLH* (*CpbHLH020*, *CpbHLH053* and *CpbHLH062*) mRNAs were reduced more than 2-fold in salt (200 mM Nacl) treated papaya seedlings. Under drought stress (25% PEG), 8 (*CpbHLH011*, *CpbHLH022*, *CpbHLH027*, *CpbHLH046*, *CpbHLH050*, *CpbHLH052*, *CpbHLH056* and *CpbHLH068*) of 22 *Cp*bHLH mRNAs were upregulated, and 3 *CpbHLH* (*CpbHLH020*, *CpbHLH042* and *CpbHLH053*) mRNAs were downregulated more than 2-fold. Under ABA treatment (100 μM), 3 (*CpbHLH027*, *CpbHLH052* and *CpbHLH056*) of 22 *Cp*bHLH mRNAs were upregulated, and 5 *CpbHLH* (*CpbHLH019*, *CpbHLH020*, *CpbHLH042*, *CpbHLH053* and *CpbHLH062*) mRNAs were downregulated. Under cold stress (4 °C), there were 4 *CpbHLH* genes (*CpbHLH027*, *CpbHLH035*, *CpbHLH056* and *CpbHLH062*) whose expression increased more than 1.5-fold, and 4 *CpbHLH* (*CpbHLH046*, *CpbHLH050*, *CpbHLH052* and *CpbHLH068*) mRNAs were reduced more than 2-fold.

Interestingly, a few transcripts of *Cp*bHLH responded to all or multiple stresses. For instance, *CpbHLH056* was sensitive to all four stresses and was upregulated distinctly under the four stresses. The orthologue of *CpbHLH056* in Arabidopsis is *BEE1* (*AtbHLH044*) (Fig. 1), which has been functionally characterized in previous reports. At low temperatures,
**Table 2  Expression levels of *CpbHLH* genes under salt, drought and ABA stresses.**

| The name of *CpbHLHs* | CK | NaCl (200 mM) | PEG (25%) | ABA (100 μM) |
|---|---|---|---|---|
| *CpbHLH006* | 1 | 0.74 ± 0.13 | 0.95 ± 0.15 | 0.75 ± 0.07 |
| *CpbHLH011* | 1 | 2.45 ± 0.04 | 4.06 ± 0.12 | 1.66 ± 0.12 |
| *CpbHLH019* | 1 | 0.80 ± 0.03 | 0.64 ± 0.02 | 0.18 ± 0.02 |
| *CpbHLH020* | 1 | 0.20 ± 0.003 | 0.39 ± 0.01 | 0.22 ± 0.01 |
| *CpbHLH022* | 1 | 4.21 ± 0.03 | 3.05 ± 0.03 | 1.54 ± 0.08 |
| *CpbHLH027* | 1 | 4.49 ± 0.09 | 2.32 ± 0.06 | 2.27 ± 0.12 |
| *CpbHLH035* | 1 | 1.62 ± 0.08 | 1.79 ± 0.07 | 0.71 ± 0.02 |
| *CpbHLH037* | 1 | 0.96 ± 0.02 | 1.04 ± 0.01 | 1.92 ± 0.07 |
| *CpbHLH040* | 1 | 1.38 ± 0.04 | 0.77 ± 0.03 | 0.51 ± 0.03 |
| *CpbHLH041* | 1 | 0.55 ± 0.01 | 0.76 ± 0.03 | 0.70 ± 0.03 |
| *CpbHLH042* | 1 | 0.55 ± 0.03 | 0.33 ± 0.04 | 0.26 ± 0.03 |
| *CpbHLH046* | 1 | 0.78 ± 0.06 | 5.46 ± 0.10 | 0.53 ± 0.03 |
| *CpbHLH050* | 1 | 0.53 ± 0.04 | 11.86 ± 0.10 | 1.18 ± 0.02 |
| *CpbHLH052* | 1 | 1.64 ± 0.06 | 8.71 ± 0.06 | 4.42 ± 0.13 |
| *CpbHLH053* | 1 | 0.44 ± 0.04 | 0.31 ± 0.02 | 0.19 ± 0.01 |
| *CpbHLH056* | 1 | 7.11 ± 0.04 | 32.65 ± 0.45 | 12.22 ± 0.21 |
| *CpbHLH060* | 1 | 0.89 ± 0.07 | 0.76 ± 0.05 | 0.75 ± 0.11 |
| *CpbHLH062* | 1 | 0.38 ± 0.02 | 0.65 ± 0.02 | 0.30 ± 0.02 |
| *CpbHLH065* | 1 | 1.64 ± 0.06 | 1.04 ± 0.03 | 0.52 ± 0.05 |
| *CpbHLH068* | 1 | 1.72 ± 0.05 | 2.37 ± 0.04 | 1.39 ± 0.07 |
| *CpbHLH069* | 1 | 0.68 ± 0.07 | 0.89 ± 0.18 | 0.52 ± 0.08 |
| *CpbHLH070* | 1 | 1.11 ± 0.02 | 1.17 ± 0.07 | 1.47 ± 0.04 |

**Notes.**
Quantitative RT-PCR was used to investigate the expression levels (shown in fold change) of the *CpbHLH*s. The expression level in the control (CK) was set at 1.0. The means of three replicates of qRT-PCR and standard deviations (SD) values are shown.

BEE1 is a positive regulator of flavonoid accumulation (*Petridis et al., 2016*), which is consistent with our results. In addition, BEE1, BEE2 and BEE3 are functionally redundant positive regulators of BR (brassinosteroid) signaling, but these transcripts are repressed by ABA (*Friedrichsen et al., 2002*). However, we found the transcription of *Cp*bHLH056 was notably upregulated (>10-fold) under ABA treatment. More interestingly, *Cp*bHLH042, which is an orthologue of BEE2 (Fig. 1), was distinctly repressed by ABA (approximately 4-fold). These results suggested that *Cp*bHLH056 and *Cp*bHLH042 may provide different functionalities compared to *Arabidopsis*. Additionally, *Cp*bHLH027 was also upregulated distinctly under four stresses. In *Arabidopsis*, the orthologue of *Cp*bHLH027 is *At*bHLH116 (ICE1) (Fig. 1), which can be induced by Nacl, ABA and cold stresses, playing an important role in the cold-responsive signaling pathway via an ABA-independent pathway (*Chinnusamy et al., 2003*). There are two orthologues of *Cp*bHLH027 in rice, one ortholog is *Os*ICE2/*Os*bHLH001, is induced by salt stress, and its overexpression can enhanced the tolerance to freezing and salt stress (*Deng et al., 2017*; *Li et al., 2010*). *Os*ICE1/*Os*bHLH002 is another ortholog in rice, which is induced by chilling stress. *Os*bHLH002 can positively regulates cold signaling via targeting *Os*TPP1, which encodes a keyenzyme for trehalose

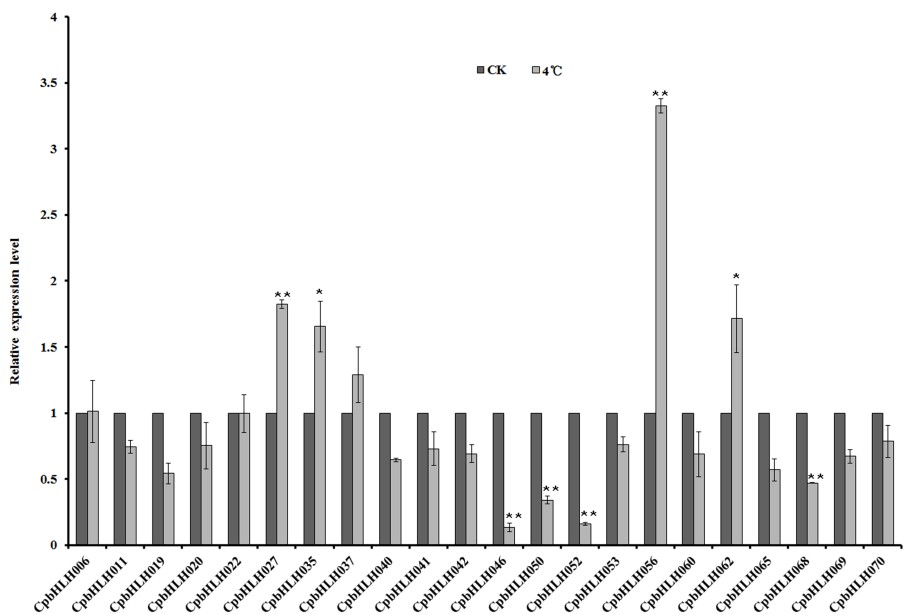

**Figure 7** **Quantitative RT-PCR analysis of 22 selected *CpbHLH* genes under cold stress condition (4 °C).** The data are expressed as means ± SD of three independent biological determinations. Untreated seedlings were used as the control groups. *$P < 0.05$ and **$P < 0.01$ (Student's *t* test) indicate significant differences between treated seedlings and control groups.

biosynthesis (*Zhang et al., 2017*). These results implied *Cp*bHLH027 plays essential roles in abiotic stresses in papaya. In addition, the transcript of *Cp*bHLH062 was also increased under cold treatment, its orthologue is *At*bHLH033/ICE2, which involving the cold response and the ABA pathway (*Fursova, Pogorelko & Tarasov, 2009*; *Kurbidaeva, Ezhova & Novokreshchenova, 2014*), implying the *Cp*bHLH062 may involved in the cold stress. *Cp* bHLH053 was downregulated under salt, drought and ABA stresses. The orthologue of *Cp*bHLH053 is *At*bHLH129 (Fig. 1), which is a transcription repressor that negatively regulates the ABA response in *Arabidopsis* (*Tian et al., 2015*), implying *Cp*bHLH053 may have the similar function with the *At*bHLH129 in the process of ABA response.

We should also noticed a few *Cp*bHLHs that showed distinct increases or decreases in their mRNA levels under different treatments, and these *Cp*bHLHs' orthologues have not been reported in previous studies. For instance, *CpbHLH050* is notably upregulated (>10-fold) under PEG treatment, *CpbHLH046* is upregulated by PEG treatment, but sharply down regulated under ABA and cold treatments, implying these genes may have additional functions than response to drought by regulating root development. *CpbHLH020* and *CpbHLH053* were downregulated (>2-fold) by Nacl, PEG and ABA stresses distinctly. We should also pay attention to these genes in the following research.

## DISCUSSION

Transcription factors (TFs) play key roles in the stress regulation network and signal pathways in plants. Basic/helix-loop-helix (bHLH) TFs are the second largest TFs family in
plants and have been identified in many species (*Cheng et al., 2018*; *Gao et al., 2017*; *Geng & Liu, 2018*; *Mao et al., 2017*; *Sun, Fan & Ling, 2015*; *Toledo-Ortiz, Huq & Quail, 2003*; *Wang et al., 2018a*; *Wang et al., 2018b*; *Zhang et al., 2018*). However, the bHLH TF family has not previously been reported in papaya (*Carica papaya* L.). In this paper, we found 73 *CpbHLH* genes in papaya. This TF family seemed to be one of the moderately sized families compared with other plant species, which might be because of the papaya has a relatively small reference genome, the size is only 372 Mb (*Ming et al., 2008*). The gene evolution changes the gene organization. In this study, we found that *CpbHLH* genes were diverse in their number introns, ranging from 0 to 10 (Data S2). This result implied these genes may have undergone numerous of genetic evolution events, and the genes in different subfamilies may have different functions (*Cheng et al., 2018*). Most *CpbHLHs* in the same subfamily shared similar gene structures and protein motifs according to the analysis of exon/intron organizations and motif compositions (Figs. 2, 3 and 4), indicating that the functions of encoded proteins in each subfamily are probably stable. However, the conserved motif analysis showed that some *Cp*bHLHs, which are mainly distributed in eight subfamilies (including V a, V b, III f, IVa, IIIb, III, I b and I a subfamilies) from the phylogenetic tree, harbor another conserved motif (motif 3) with a length of 36 amino acids (Figs. 2 and 4), indicating that these proteins may have additional functions.

Promoter *cis*-acting regulatory element analysis showed that *cis*-elements could be divided into three main categories: light responsive, abiotic and biotic stresses and plant growth and development. Especially in abiotic and biotic stresses, the most common response elements were ABA (29.15%), MeJA (23.62%) and SA (9.78%), suggesting that these phytohormones may play important roles in the regulation of papaya growth and development (Fig. 5 and Data S4). In addition, the promoter *cis*-acting element involved in the abscisic acid responsiveness analysis is consistent with the qRT-PCR results (Data S4 and Table 2), showing four genes (*CpHLH020/-027/-053/-056*) involved in abscisic acid response. Another two genes (*CpHLH020/-062*) were also identified that were involved in abscisic acid response by GO enrichment analysis and qRT-PCR (Data S6 and Table 2). We also identified a large number of *cis*-acting elements in *CpbHLH* genes that may respond to drought (MBS, 8.67%), which is also consistent with the qRT-PCR (including seven genes: *CpbHLH027/-050/-056/-011/-068/-042/-053*). Other genes also had important elements, including LTR, TC-rich and WUN-motifs, which indicated plant responses to low temperatures, defense stresses and wound-responsiveness, respectively. These results implied *CpbHLH* genes may have a wide range of functions in papaya growth, disease resistance, and response to environmental conditions. We also analyze the *cis*-elements in the promoter regions of putative orthologous genes in *Arabidopsis* (Fig. S1 and Data S4). And we compared the promoter cis-elements of *bHLH* genes in *Arabidopsis*, papaya and previously reported bamboo (*Cheng et al., 2018*). The result showed that the promoter *cis*- elements of *bHLH* genes in these three plants were divided into three categories, and most of elements were the same. These similar results implied that most of the promoter *cis*-elements of bHLH family were conserved in *Arabidopsis*, papaya and bamboo. The most notable is, the percentage of MeJA responsive *cis*-elements in papaya and *Arabidopsis* (23.62%, 30.61%) were less than bamboo (43.39%). And the
percentage of SA responsive *cis*-elements in papaya and *Arabidopsis* (9.78%, 5.87%) were also less than bamboo (10.31%). SA is a phytohormone that plays important roles in plant defenses against pathogens (*Pokotylo, Kravets & Ruelland, 2019*). MeJA also has been identified as a vital cellular regulator that mediates defense processes (*Cheong & Choi, 2003*). So, the stress-responsive elements in papaya and *Arabidopsis* were corresponding less than bamboo, including drought-responsive elements, wound-responsive elements, low temperature-responsive elements, and defense and stress responsive elements. These results may help explain why papaya is more sensitive to external stresses compared to bamboo.

Many studies have shown that *bHLH* genes are involved in various abiotic and biotic stresses responses. We randomly selected 22 genes to investigate their expression profiles by using qRT-PCR under salt, drought, ABA and cold stresses (Fig. 7 and Table 2). The results revealed some candidate *CpbHLH* genes that might be responsible for abiotic stress responses in papaya. For example, *Cp*bHLH027, *Cp*bHLH062, *At*bHLH116 (ICE1), *At*bHLH33 (ICE2), *Os*bHLH001 (*Os*ICE2) and *Os*bHLH002 (*Os*ICE1) were clustered within one clade. Among them, *At*bHLH116 (ICE1), *At*bHLH33 (ICE2), *Os*bHLH001 (*Os*ICE2) and *Os*bHLH002 (*Os*ICE1) have been reported function in chilling stress in *Arabidopsis* and rice (*Chinnusamy et al., 2003*; *Deng et al., 2017*; *Fursova, Pogorelko & Tarasov, 2009*; *Li et al., 2010*; *Zhang et al., 2017*). And the transcripts of *Cp*bHLH027, *Cp*bHLH062 were increased under chilling stress in this study, implying that *Cp*bHLH027 and *Cp*bHLH062 may be also involved in the process of chilling stress. The orthologue of *CpHLH056* in *Arabidopsis* is *BEE1* (*At*1G18400), which is a positive regulator of flavonoid accumulation (*Petridis et al., 2016*). BEE1, BEE2 and BEE3 are functionally redundant positive regulators of BR signaling, and their transcription is repressed by ABA in *Arabidopsis* (*Friedrichsen et al., 2002*). However, we found that the transcription of *CpbHLH056* is notably upregulated (> 10-fold) under ABA treatment rather than downregulated in this study. These results imply that *CpbHLH056* may involved in the process of ABA stress but has different function compared to *Arabidopsis*. We have also noticed a few candidate *Cp*bHLHs that showed distinct increases or decreases in their mRNA levels under different treatments, and these *Cp*bHLHs' orthologues have not been reported in other plants. For instance, *Cp*bHLH050, *Cp*bHLH020, *Cp*bHLH046, *Cp*bHLH053, and so on. These findings provide important candidate genes/proteins necessary for further functional research on the bHLH family in papaya.

## CONCLUSIONS

In conclusion, the study performed a genome-wide analysis of basic helix-loop-helix (bHLH) transcription factors in papaya. As a result, a total of 73 *bHLH* genes were identified in papaya, and these *CpbHLHs* were classified into 18 subfamilies with one orphan, which was consistent with the earlier results showing that the bHLH subfamily in plants can be divided into 15–25 subfamilies. Almost all of the *CpbHLHs* in the same subfamily shared similar gene structures and protein motifs according to analysis of exon/intron organizations and motif compositions. These results further supported the

classification predicted by the phylogenetic tree. Compared to rice and *Arabidopsis*, the amino acid sequences of the *Cp*bHLH domains were quite conservative, especially Leu-27 and Leu-63. Promoter *cis*-element and GO annotation analysis revealed that most of the *CpbHLHs* could respond to various biotic/abiotic stress-related events. Abiotic stress treatment and quantitative real-time PCR (qRT-PCR) assay further supported promoter *cis*-acting regulatory element and GO annotation analysis, revealed some important candidate *CpbHLHs* that might be responsible for abiotic stress responses in papaya. We completed the first comprehensive genome-wide analysis of the *bHLH* gene family in papaya, and our results provide information necessary for further functional research of the bHLH family in papaya.

### Funding

This work was supported by the Presidential Foundation of Guangdong Academy of Agricultural Sciences, China (No. 201820), the Key-Area Research and Development Program of Guangdong Province (No. 2019B020214005), and the Innovation Team Program of Modern Agricultural Industry Technology System in Guangdong Province of China (2019KJ116). The funders had no role in study design, data collection and analysis, decision to publish, or preparation of the manuscript.

### Grant Disclosures

The following grant information was disclosed by the authors:
Presidential Foundation of Guangdong Academy of Agricultural Sciences, China: 201820.
Key-Area Research and Development Program of Guangdong Province, China: 2019B020214005.
Innovation Team Program of Modern Agricultural Industry Technology System in Guangdong Province, China: 2019KJ116.

### Competing Interests

The authors declare there are no competing interests.

### Author Contributions

- Min Yang conceived and designed the experiments, performed the experiments, analyzed the data, prepared figures and/or tables, authored or reviewed drafts of the paper, and approved the final draft.
- Chenping Zhou and Hu Yang performed the experiments, analyzed the data, prepared figures and/or tables, authored or reviewed drafts of the paper, and approved the final draft.
- Ruibin Kuang and Bingxiong Huang performed the experiments, analyzed the data, authored or reviewed drafts of the paper, and approved the final draft.
- Yuerong Wei conceived and designed the experiments, authored or reviewed drafts of the paper, and approved the final draft.

## Data Availability

The raw data are available in the Supplementary Files.

## Supplemental Information

Supplemental information for this article can be found online at http://dx.doi.org/10.7717/peerj.9319#supplemental-information.

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
