# Peer review of "Genome-wide analysis of basic helix-loop-helix transcription factors in papaya (Carica papaya L.)"

_PeerJ, doi:10.7717/peerj.9319_

## Round 0.1 · original submission · Minor Revisions

The reviewers have high estimates for the manuscript. However there some minor remarks, including the remarks on the figure quality and the discussion section on bHLH genes. Waiting your revised manuscript.

Reviewer 1 ·

Basic reporting

The article should include sufficient introduction and background to demonstrate how the work fits into the broader field of knowledge. Relevant prior literature should be appropriately referenced.

Experimental design

Methods described with sufficient detail & information to replicate.

Validity of the findings

The data on which the conclusions are based must be provided or made available in an acceptable discipline-specific repository. The data should be robust, statistically sound, and controlled.

Reviewer 2 ·

Basic reporting

The manuscript entitled ‘Genome-wide analysis of basic helix-loop-helix transcription
factors in papaya ( Carica papaya L. )’ describes genome-wide identification of Carica papaya bHLH transcription factors downloaded from the Plant TFDB V4.0 database. The bHLH transcription factors were then analyzed bioimformatically, and their expression patterns after abiotic stress treatment by qRT-PCR. Together, the authors draw to conclusion that some important candidate CpbHLHs are be responsible for abiotic stress responses in papaya.

In general, the manuscript is well written, contains some fairly-novel results, and is appropriate for PeerJ.

Experimental design

no comment

Validity of the findings

no comment

Additional comments

The manuscript entitled ‘Genome-wide analysis of basic helix-loop-helix transcription
factors in papaya ( Carica papaya L. )’ describes genome-wide identification of Carica papaya bHLH transcription factors downloaded from the Plant TFDB V4.0 database. The bHLH transcription factors were then analyzed bioimformatically, and their expression patterns after abiotic stress treatment by qRT-PCR. Together, the authors draw to conclusion that some important candidate CpbHLHs are be responsible for abiotic stress responses in papaya.

In general, the manuscript is well written, contains some fairly-novel results, and is appropriate for PeerJ. I have only a few minor comments:

a.In the Background section, the authors introduced the background of papaya and gene family analysis in papaya only in two sentences. The authors should rewritten this part and cite more references such as below for reference citation.

Pan LJ and Jiang L. Identification and Expression of the WRKY Transcription Factors of Carica Papaya in Response to Abiotic and Biotic Stresses. Mol Biol Rep, 41 (3), 1215-1225.
Liu KD, Yuan CC, Feng SX, et al. Genome-wide analysis and characterization of Aux/IAA family genes related to fruit ripening in papaya (Carica papaya L.). BMC Genomics, 2017, 18:351.
Liu KD, Yuan CC, Li HL, et al. Genome-wide identification and characterization of auxin response factor (ARF) family genes related to flower and fruit development in papaya (Carica papaya L.). BMC Genomics, 2015, 16: 901.

b.Moreover, the authors should also show the cis-motifs in the promoter regions of putative orthologs in Arabidopsis genes to provide comparative aspects in the promoter regions as well as gene expression in response to abiotic stress treatments in the Figure 5 and 7.

c.The resolution in figure 4 is very lower, should be revised.

·

Basic reporting

Author used clear and unambiguous, professional English used throughout. They cited appropriate previous studies.

Experimental design

Research question well defined, relevant & meaningful. It is stated how research fills an identified knowledge gap. They performed detailed analysis in terms of bHLH in papaya genome.

Validity of the findings

They discussed most of the important findings in their results. Conclusions are well stated, linked to original research question & limited to supporting results.

Additional comments

Revise page 8, line 156.

Please use same style for gene and protein name.

How did you select stress conditions? Do you have any reference study?

How did you determine the genes studied in gene expression analysis with qRT-PCR?

Please discuss why most of bHLH genes gave responses to the all stress conditions?

---

## Round 0.2 · accepted · Accept

Thank you for the detailed answer to the reviewers. Since some reviewers can’t reply in time, I checked the manuscript updates based on the previous comments. I believe this interesting work in plant science should be published now.

Reviewer 2 ·

Basic reporting

No comment.

Experimental design

No comment.

Validity of the findings

No comment.

Additional comments

The revised manuscript has been much improved and answered all my concerns. I have no more comments. I think the revised manuscript is suitable for publication in PeerJ.